# Nursing Home Residents Hospitalization at the End of Life: Experience and Predictors in Portuguese Nursing Homes

**DOI:** 10.3390/ijerph20020947

**Published:** 2023-01-04

**Authors:** Helena Bárrios, José Pedro Nunes, João Paulo Araújo Teixeira, Guilhermina Rego

**Affiliations:** 1Hospital do Mar Cuidados Especializados Lisboa, 2695-458 Bobadela, Portugal; 2Faculty of Medicine, University of Porto, 4200-319 Porto, Portugal

**Keywords:** end-of-life care, care transitions, nursing home, palliative care, geriatrics

## Abstract

(1) Background: Nursing Home (NH) residents are a population with health and social vulnerabilities, for whom emergency department visits or hospitalization near the end of life can be considered a marker of healthcare aggressiveness. With the present study, we intend to identify and characterize acute care transitions in the last year of life in Portuguese NH residents, to characterize care integration between the different care levels, and identify predictors of death at hospital and potentially burdensome transitions; (2) Methods: a retrospective after-death study was performed, covering 18 months prior to the emergence of the COVID-19 pandemic, in a nationwide sample of Portuguese NH with 614 residents; (3) Results: 176 deceased patients were included. More than half of NH residents died at hospital. One-third experienced a potentially burdensome care transition in the last 3 days of life, and 48.3% in the last 90 days. Younger age and higher technical staff support were associated with death at hospital and a higher likelihood of burdensome transitions in the last year of life, and Palliative Care team support with less. Advanced Care planning was almost absent; (4) Conclusions: The studied population was frail and old without advance directives in place, and subject to frequent hospitalization and potentially burdensome transitions near the end of life. Unlike other studies, staff provisioning did not improve the outcomes. The results may be related to a low social and professional awareness of Palliative Care and warrant further study.

## 1. Introduction

With the progressive aging of the population, there is an increasing number of frail and dependent elderly, with multiple health and social vulnerabilities [1,2,3]. Frailty is a condition characterized by functional decline and increased vulnerability to stressors [4,5]. Its prevalence increases with age. In a recent meta-analysis of population-level studies in 62 countries, O’Caoimh et al. found a 16% prevalence of frailty in studies with inclusion age of 60–69 years and a 51% prevalence in the 90+ cohort [6]. Institutional residential care in NH is one of the options for meeting the care needs of the frail elderly, and an increase in the demand for these services is expected [7,8]. However, the healthcare coverage in most institutions is still uneven. The adaptation of each NH to the needs of its residents is ongoing, with some of them including indoor medical, nursing, and therapy support, but the regulatory framework in most countries is outdated, and a lack of appropriate financial support is widespread [9].

Some of the possible consequences of the lack of appropriate healthcare coverage are the under-recognition of palliative care needs and the overuse of emergency department visits by the residents. Both factors are interconnected and might impact the quality of life of the residents and represent a significant burden on healthcare systems [10].

NH are considered by most residents as their home and desired place of death [11,12]. However, the end-of-life care needs of the residents can be challenging for teams with low healthcare support [13]. In this context, healthcare transitions, mostly emergency department visits and hospitalization, are frequent among the institutionalized elderly [11,14].

For older people with dementia, emergency department visits or hospitalization near the end-of-life are considered, population-wise, a marker of healthcare aggressiveness [15]. For the patient, the family and the care team, care transitions near the end-of-life may be disruptive and are often associated with healthcare-associated infections, pressure ulcers, and worsening mental, functional, and behavioral status [16]. However, care transitions may be necessary for meeting care needs, including symptom control and the treatment of fractures, and may improve the residents´ quality of life [17]. To address this issue, the concept of burdensome transition has been developed. It was first introduced in 2009 by Mitchell, et al., in a prospective cohort study on the clinical course of advanced dementia. In this study, hospitalization was considered one of the possible “burdensome interventions” suffered by the residents [18]. The first large-scale study on burdensome transitions was published by Gozalo, et al., in 2011. In this study, data from the Medicare Minimum Data Set of 474,829 NH decedents were analyzed. The definition of burdensome transition was based on previous qualitative data and expert opinion. Three types of transitions were classified as being potentially burdensome: any transfer in the last 3 days of life, a lack of continuity of nursing home facilities before and after a hospitalization in the last 90 days of life (i.e., going from nursing home A to the hospital and then to nursing home B), and multiple hospitalizations in the last 90 days of life [19]. The concept of burdensome transition continues to be investigated and is evolving [20,21,22]. [19,20,21,22] In a recent scoping review, definitions of burdensome transitions were characterized by the following features: transition setting trajectory, number of transitions, temporal relationship to end-of-life, and quality of transitions [23]. Most studies included in the review were based on the original work by Gozalo, et al.

National and local resources, different cultural and social backgrounds, and personal views and attitudes about death will implicate different pathways in end-of-life care, including care transitions [24]. In this context, national studies are needed in order to better understand care needs and guide policies and practices.

Portugal is a Western European country with a life expectancy above the average of the Organization for Economic Co-operation and Development (OECD) countries but low quality of life at older ages, with 85.2% of older adults, rating their own health as fair, poor, or very poor, and 64% living with limitations in daily activities [25]. In a study based on the Survey of Health, Ageing and Retirement in Europe, Portugal had the highest frailty prevalence in older adults (15.6% vs. 7.7% mean frailty prevalence in 18 countries) [3].

Death at hospital is also above the OECD average, with 59.9% of deaths in 2019 occurring in hospitals [26]. Unlike other countries, there is a trend for an increase in hospital deaths from patients dying from conditions needing palliative care [27,28].

In Portugal, there is a paucity of care home-based end-of-life care studies, and to our best knowledge, no studies analyzing the nursing home residents´ healthcare transitions in end-of-life have been reported. Given the chronically overused emergency departments, and low capacity of intermediate care responses (National Continuity of Care Network), investigating care home capacity to deliver good end-of-life care is particularly relevant [29].

With the present study, the authors aim to identify and characterize hospitalization in the last year of life of Portuguese NH residents, characterize care integration between the NH and the hospital and identify predictors of death at hospital and potentially burdensome transitions.

## 2. Materials and Methods

The study was designed to address the research questions: (1) what is the scope of the problem of hospitalization at the end-of-life of Portuguese NH residents? (2) what are the predictors of death at hospital and potentially burdensome transitions of NH residents at the end of life? As potential predictors, sociodemographic and clinical characteristics of the NH residents, and NH characteristics will be considered. The quality of healthcare transitions will be evaluated and discussed.

### 2.1. Study Design

A retrospective after-death study was performed, covering the period from September 2018 to February 2020. Given the changes in morbidity and mortality in care homes caused by the COVID-19 pandemic (that started in March 2020 in Portugal), the pandemic period was excluded from this study in order to minimize bias.

#### Study Population

The study included deceased residents from a sample of Portuguese NH belonging to charities affiliated with União das Misericórdias Portuguesas (UMP). UMP is a collective entity that represents the Portuguese religious not-for-profit organizations and gives technical and administrative support, enabling a better uniformization of institution characteristics. There are in total 387 NH with nationwide implementation. These institutions had 27,455 residents in the year 2020. The mortality rate was 6.74% (749 deaths within the NH, 1102 in the hospital). For the calculation of a significant sample of deceased patients in an 18-month period, at a confidence level of 95%, margin of error of 5%, population proportion of 10%, and total population of 27,455, the minimal sample size of deceased patients was calculated at 138. A convenience sampling method, stratified by geographical region was conducted. Cases were included if death occurred within the study period, and residents were living in the care home for at least 90 days prior to death, in line with previous studies [30,31,32,33].

### 2.2. Variables and Instruments

Variables related to the resident and institutional characteristics were studied. Deceased resident data was obtained through analyses of the administrative and clinical records, death certificates, and hospital discharge information.

The outcome variables considered were place of death and two kinds of potentially burdensome transitions, as originally described by Gozalo, et al.: any transfer in the last 3 days of life and multiple hospitalizations in the last 90 days of life (defined as either more than two hospitalizations for any reason or more than one hospitalization for pneumonia, urinary tract infection, dehydration, or sepsis). Fewer hospitalizations were required for these diagnoses because they are predictable end-of-life events in patients with advanced cognitive impairment and are potentially manageable with appropriate advance care planning, without the need for hospitalization [19]. To obtain data on the outcome variables, the national death certificate electronic registry, National Health Service clinical record, NH clinical record, and written hospital discharge information were consulted.

The predictor variables studied and respective source were: age at time of death; gender; civil status; schooling years; religion; annual income; length of stay in the care home (administrative records); advanced directives; performance status on the month prior to death (Eastern Cooperative Oncology Group -Performance status) [34], chronic health diagnosis as described by the attending physicians (NH clinical records and National Health System electronic clinical records); cause of death (national death certificate electronic registry); [34] number of residents in the NH, health support from Palliative Care team; belonging to a charity with a Long Term Care facility (LTCF) (national government electronic database), number of deceased residents in the study period; composition and hour coverage by the technical staff; and existence of advance care planning protocols (namely, advance care planning discussion with the residents and families, do not resuscitate and do not transfer orders) (interview with the director of the institution).

Data on healthcare transitions regarding the 12 months prior to death were collected for descriptive analyses: number of hospital admissions; days spent at hospital; clinical referral information; written clinical information at readmission to the care home after hospitalization; motif for referral; discharge diagnoses; health hazards during hospital stay. These data were obtained by consulting NH clinical records, printed hospital discharge information, and National Health System clinical records.

### 2.3. Statistical Analysis

Sample characterization was performed using descriptive analyses. To identify predictors of dying at hospital, and potentially burdensome transitions, the adjusted odds ratio (OR) of death at hospital, was calculated using binary logistic regression in continuous and ordinal variables, and χ2 or Fisher’s exact test in dichotomous variables. All reported *p*-values are two-sided with significance defined as *p* < 0.05. All analyses were performed with SPSS 28.01.0 for Windows (IBM Corporation, Armonk, NY, USA).

## 3. Results

The study included a sample of 7 NH, with nationwide distribution, the institutions sheltered in total 614 residents, giving a mean of 93 residents per NH (min 60, max 120). In the 18-month study period, a total of 206 residents died and the mean number of deceased patients in each NH was 29 (min 17, max 43). All of the NH had in-doors medical support. Only one of the NH had protocols for advanced care planning and do-not-resuscitate order discussion. No protocols for do-not-transfer orders were in place in any of the studied nursing homes.

### 3.1. Residents

A total of 206 NH deceased residents were analyzed for eligibility. A total of 30 were excluded due to death within the 90 days following admission. Thus, 176 deceased residents were included in the study (Table 1). The majority were females (69.9%), widowed (54.5%), and had close family relatives (68.8%). The sample had low education (mean 3.2 years), low income (7092 Euro/year), and high age at time of death (mean age at time of death 86 years (SD 8.6)). The clinical characteristics reveal high dependency and multi-morbidity: a four ECOG PS was present in 69.9% of the deceased residents in the month prior to death, and the mean number of chronic diseases was 3.5 (SD 1.5). Dementia (from all causes) was identified in 86.9 % of the deceased residents. 97.2% of the deceased residents were Catholic. Only 2.3% had formal advanced directives in place at time of death. Death was attributed to infectious causes in 45% of cases, and neurological and cardiac disease were the most frequent organ disease diagnoses.

### 3.2. Hospitalizations and In-Hospital Death

52.3% of NH residents’ deaths occurred at hospital (Table 2). 82.4% of residents had at least one acute care referral in the last year of life in the institution, and 44.9% had two or more. Care transitions fulfilled Gozalo´s criteria for burdensome transitions [19] in 59 (33.5%) of the residents, due to transitions in the last 3 days of life, and 85 (48.3%) residents, due to multiple transitions in the last 90 days. A total of 288 referrals occurred in the study population. In the majority of cases (50.8%), referrals resulted in a hospital stay of 1 night or less—included in this number are deaths on the day of referral (23 cases, 8.2% of referrals). A total of 36.1% of the referrals resulted in hospitalizations of 2 to 10 days, and only 13% of cases had longer hospitalizations.

Health hazards, mainly pressure ulcers and health-associated infections, were described only in longer stays, and the rate was overall low (in 8.3% of the hospitalizations). Information was sent from the NH to the hospital in almost all cases, through three possible means: written information, via the emergency personnel, or through accompanying NH staff. Written return information accompanying the resident was received in the nursing home in 89% of the cases. Acute care use was more frequent near death, with 45.6% of referrals occurring in the last month of life (Figure 1).

The reasons for referral were mostly dyspnea and other respiratory symptoms (39.2%), followed by focal neurological symptoms (15.9%) and falls (10.6%). The discharge diagnosis was infection in 43.1% of cases (including respiratory, urinary tract infections, and sepsis), followed by neurological disease (13.9%) and trauma (10.1%).

### 3.3. Bivariate Analysis

Residents over 85 years old had lower odds of dying at hospital or having burdensome transitions (OR = 0.45 (CI = 0.24–0.84) for dying at hospital; OR = 0.51 (CI = 0.27–0.97) for care transitions in the last 3 days of life; OR = 0.36 (CI = 0.18–0.65) for multiple transitions in the last 90 days of life (Table 3). No other demographic or clinical variable was associated with the outcomes, except for residents with chronic lung disease, which had an OR = 4.95 (CI = 1.76–13.95) for multiple care transitions in the last 90 days of life. The resident was more likely to die at the NH if neurological or cardiac disease was the declared cause of death, and more likely to die at hospital if infection was the terminal event. Facility characteristics predicted death at hospital, and potentially burdensome transitions in the last three days of life, with higher staff hours (medical, nurse, and professional caregiver) implying higher odds of death at hospital and care transitions in the last three days of life. Higher nursing staff coverage was also associated with higher odds of multiple care transitions in the last 90 days. The support of an external Palliative Care Team lowered the odds of death at hospital (OR = 0.24; CI = 0.11–0.54), and care transitions in the last 3 days (OR = 0.37; CI = 0.15–0.91), but not of multiple care transitions in the last 90 days (OR = 0.48; CI = 0.22–1.00).

## 4. Discussion

With the present study in a sample of Portuguese nursing home residents, we found a high proportion of death at hospital (52.3% of the residents), acute care referrals in the last year of life (82.4%), and potentially burdensome transitions (33.5% of the residents having been transferred in the last 3 days of life and 48.3% in the last 90 days of life). In a systematic review of 35 studies, Allers et al., describe a median of 22.6% of in-hospital deaths [14], Gozalo et al. found at least one potentially burdensome transition in 19% of the deceased residents, in a sample of US NH [19]. Younger age at time of death, COPD diagnosis, and higher technical staff coverage increased the odds of acute care referral and death at hospital.

The age distribution and its influence on the outcomes identified is similar to other studies [19,35,36,37]. Residents died at an old age, with a very high proportion of multi-morbidity and cognitive decline. Younger residents (85 years or less) were more likely to be transferred to acute care. These results are also described in other studies and suggest a more aggressive approach near end-of-life in this group [19,30,31,32,37]. Diagnostic and prognostic uncertainty by the healthcare professionals at the NH [38], very low advance directive discussion [20], and family and social pressure favoring hospital care as the best care [28] may contribute to the explanation of these findings.

The education level and income of the residents are very low in this sample, reflecting the socio-demographic characteristics of the older population in Portugal and specifically of the residents in the studied charities [26].

We found a lower prevalence of advanced directives than studies in other countries [21,39]. It is, however, much higher than the general Portuguese population (with only 0.07% of adult citizens having an advanced directive registered in the national advance directives database in 2019). Also, when in place, the advanced directives concerned mainly the refusal of blood transfusions, due to religious beliefs, and not end-of-life care as a whole. This may reflect a low social awareness regarding end-of-life issues and palliative care in a country lacking effective integrated palliative care support [28]. Setting the goals for end-of-life can be done without the formalization of advance directives, and, at the same time, advance care planning is only one of the steps of decision-making at the end of life [20]. However, the absence of defined protocols for advance directive discussion, including do-not-resuscitate and do-not-transfer orders may have contributed to the high rate of transferals and in-hospital death identified in this sample.

The rate of hospital deaths varies widely across different studies with a range of 5.9% to 77.1% [14]. Even though this outcome is related to multiple complex factors, a national trend can be found, with Japanese studies portraying consistently higher rates of hospital deaths (46.9–77.1%). These results outline the social and cultural background that influences residents, families, and professionals, as a possible important factor to consider. In Portugal, population-wise, most deaths including those caused by conditions needing palliative care take place in hospital (61.9%), with an increasing trend over the years. Also, as described by Gomes et al., unlike in other countries, in the general population, a cancer diagnosis increases the odds of dying at hospital. This suggests a reliance on hospitals as providers of the best available care, and low palliative care awareness or preparedness [28]. This is in line with the results of the present study, where the time distribution of hospital referrals along the last year of life reveals a strong shift to the day of death, with most of the transferals occurring in the last days of life.

Unlike other studies, in the present sample, higher technical staff provisioning was associated with higher rates of resident referral to acute care and in-hospital death. Although there is robust evidence that many acute care referrals from NH are potentially avoidable, it may be difficult to identify avoidability at the time of transferal. Symptoms are only weakly predictive of hospital diagnoses, and prognostication may be a difficult endeavor [38]. NH residents as described in the present study are frail, dependent, and have multi-morbidity and cognitive impairment. Additionally, advance care planning did not take place in most of the cases, and the social and cultural background in Portugal favors hospital care as the optimal care [28]. In this context, acute care referral may be expected if there is a real or perceived inability of NH staff to manage acute conditions in seriously ill residents [40,41]. In the present study, respiratory symptoms were the most frequent motif for referral decisions and evidence this struggle to deal with potentially distressing symptoms in the NH.

Infection was the most frequent discharge diagnosis and a predictor of death at hospital. In this very frail population, infection is expected to be a terminal event, the implementation of end-of-life care discussions and advanced directives might allow for the mitigation of death at hospital in these circumstances [12].

The support of palliative care teams improved the outcomes, with less hospitalization and in-hospital deaths. External palliative care team support has not been extensively studied, since most studies come from countries with strong hospice care, with Hospice/NH partnerships being the most studied care model [41,42]. External palliative care team support and facility-based teams and hospice units are alternative models of care [42]. Palliative Care consultations have proved beneficial and may be particularly useful in contexts with low in-patient Palliative Care unit availability [21], a context similar to the one found in Portugal.

Clinical information was available throughout the patient’s journey, flowing between the Hospital and the NH. The results reinforce the good organization of the NH and the adequacy of existing care transition protocols at acute care hospitals. It is unlikely that the high proportion of care transitions was related to the quality of care transitions [43].

Strengths and limitations:

This is, to our knowledge, the first study on this subject in Portugal. Although the sample size is small, it represents a significant sample of a large subset of Portuguese NH, and the included NH have nationwide distribution. The study design was retrospective, but this is a common feature of mortality studies [14,44,45]. The main conclusions are related to features not singly related to chart review, namely institution characteristics, death data (that were obtained from a national electronic death certificate database), and transferal information obtained from linking administrative and clinical data. Some effects might, however, have been under-recognized, namely regarding the clinical characteristics of the residents. The low schooling and low income presented a floor effect and did not allow the study of the impact of these variables on the outcome. Part of the information on the facility characteristics was not available on a national database and was obtained through interviews, which may introduce some bias.

Further studies are warranted to better understand results not coincident with previous research, namely higher technical staffing being predictive of more aggressive end-of-life care. Additional country-specific quantitative and qualitative studies addressing the beliefs and knowledge regarding end-of-life care in the NH may give a valuable contribution to understanding resident hospitalization at the end of life. Improved knowledge on this subject may guide policymakers in the implementation of successful measures to mitigate end-of-life care aggressiveness in the Portuguese NH. The results of the present study point to the need to address the technical teams and promote Palliative Care support to the NH.

## 5. Conclusions

This study characterized end-of-life transferals and their predictors in a sample of Portuguese NH. The residents included were old and frail, with a high proportion of dementia and dependency, and without advanced directives in place. Death at hospital, acute care referrals in the last year of life, and potentially burdensome transitions were very frequent and most transferals occurred near the end of life. Being younger, suffering lung disease, and dying from infection were predictors of negative outcomes. Support from an external Palliative Care team favored death at the NH. Unlike previous studies, higher technical staff provisioning at the NH predicted more transferals and death at hospital. However, other quality indicators such as information continuity, duration of hospital stays, and health hazards during hospital stays were globally positive.

The results may reveal a low End-of-Life planning/Palliative Care awareness (social, residents/family, professionals), with a shift towards hospital as the best place for the care of the dying frail and warrants further study. A need was identified to promote the improvement of end-of-life care in Portuguese NH. Increasing country-specific scientific knowledge on end-of-life care, may inform policymakers and allow for the implementation of measures that foster the best care at the best place, in order to improve the quality of life and quality of death of NH residents.

## Figures and Tables

**Figure 1 ijerph-20-00947-f001:**
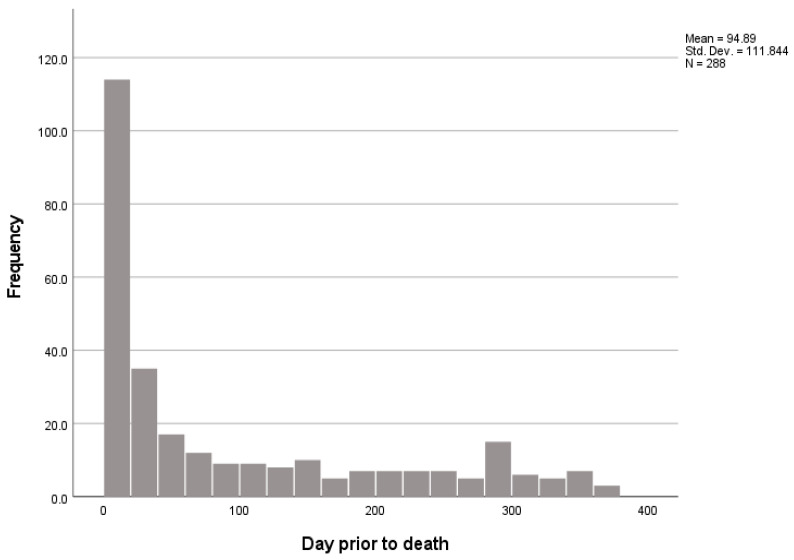
Histogram of frequency of referral to acute care, related to days prior to death. N—number; Std. Dev.—standard deviation.

**Table 1 ijerph-20-00947-t001:** Socio-demographic and clinical characteristics of the NH Residents. N—number; SD—standard deviation.

Variable		Value
Residents, N		176
Socio-demographic characteristics		
Gender, N (%)	Male	53 (30.1)
	Female	123 (69.9)
Age at death; years, mean ± SD		85.6 ± 8.6
Length of stay; months, mean ± SD		52.6 ± 46.0
Civil status, N (%)	Single Married Divorced Widowed	19 (10.8) 46 (26.1) 14 (8.0) 96 (54.5)
Family, N (%)	No family	32 (18.5)
	1st degree	119 (68.8)
	2nd degree	22 (12.5)
Schooling, years, mean ± SD		3.5 ± 2.6
Annual income, Euro, mean ± SD		7023.9 ± 4178.0
Clinical Characteristics		
Performance Status, N (%)	2	7 (4.0)
	3	46 (26.1)
	4	123 (69.9)
Number of chronic diseases, N mean ± SD		3.5±1.5
Chronic diseases, N (%)	Dementia	153 (86.9)
	Arterial Hypertension	121 (68.8)
	Heart Disease	76 (43.2)
	Diabetes mellitus	56 (31.8)
	Stroke	41 (23.3)
	Chronic Renal Disease	32 (18.2)
	Lung Disease	24 (13.6)
	Cancer	24 (13.6)
Cause of death, N (%)	Infection Neurological Cardiac Cancer Other	80 (45.5) 29 (16.5) 21 (11.9) 15 (8.5) 31 (17.6)

**Table 2 ijerph-20-00947-t002:** Hospitalization in the last year of life. N—number.

Variable	Value
Referrals to Acute Care in the Last Year of Life, N	288
Referrals to Acute Care in the last year per resident, N (%)	
0 1 ≥2	31 (17.6) 66 (37.5) 79 (44.8)
Number of nights at hospital per episode, N (%)	
0–1 2–10 >10	148 (51.4) 104 (36.1) 36 (12.5)
Days prior to death	
At day of death 1–3 days 4–30 days 31–90 days >90 days	26 (8.2) 38 (11.9) 69 (21.6) 51 (16.0) 104 (32.6)
Reason for referral, N (%)	
Dyspnea and other respiratory symptoms Focal neurological symptoms Falls Bleeding (all causes) Altered consciousness Other	111 (39.2) 45 (15.9) 30 (10.6) 24 (8.5) 19 (6.7) 54 (19.1)
Discharge diagnosis, N (%)	
Infection Neurological Trauma Cardiac Cancer Other	124 (43.1) 40 (13.9) 29 (10.1) 21 (7.3) 19 (6.6) 55 (19.1)
Care integration quality indicators	
Referral information Discharge information Health hazards	88% 95% 24 (8.3%)

**Table 3 ijerph-20-00947-t003:** Predictors of death at Hospital and potentially Burdensome Transitions. (OR, OR (95% CI) if *p* < 0.05). N—number; Ref—reference, LTCF—Institution with Long Term Care Facility.

	Death at Hospital	Burdensome Transition Last 3 Days	Burdensome Transition Last 90 Days
Residents, N (%)	92 (52.3)	59 (33.5)	85 (48.3)
Socio-demographic Variables			
Gender Male (ref. Female)	0.75	0.91	0.75
Age at death >85 years (ref. ≤85)	0.45 (0.24–0.84)	0.51 (0.27–0.97)	0.36 (0.18–0.65)
Length of stay	1.00	0.83	1.00
Civil status Single Married Divorced Widowed	ref 1.07 0.66 1.00	ref 1.53 0.59 0.97	ref 3.08 2.89 1.80
Family No family 1st degree 2nd degree	ref 0.72 0.68	ref 0.63 0.43	ref 0.74 0.47
Schooling	1.02	1.06	1.09
Annual income	1.00	1.00	1.00
Clinical Variables			
Performance Status 2 3 4	Ref 0.34 0.14		
Number of chronic diseases	1.15	0.77	1.04
Chronic diseases Dementia Arterial Hypertension Heart Disease Diabetes mellitus Stroke Chronic Renal Disease Lung Disease Cancer	0.82 0.97 1.49 1.48 1.22 1.67 2.00 0.61	1.50 1.05 0.86 1.15 1.37 1.24 1.83 1.50	1.54 0.86 1.03 0.90 1.03 1.27 4.95 (1.76–13.95) 0.60
Cause of death, N (%) Infection Neurological disease Cardiac disease Cancer Lung disease Other	2.36 (1.28–4.34) 0.42 (0.18–0.96) 0.18 (0.06–0.56) 0.58 2.36 5.26 (1.47–18.88)		
Facility Characteristics			
N residents	1.02 (1.01–1.04)	1.02 (1.01–1.04)	1.02 (1.00–1.03)
Physician/resident ratio	5.16 (1.74–15.30)	6.56 (1.86–23.12)	1.26
Nurse/resident ratio	3.61 (1.86–6.99)	2.09 (1.05–4.16)	3.54 (1.83–6.88)
Caregiver/resident ratio	1.33 (1.12–1.58)	1.35 (1.11–1.64)	1.15
LTCF	0.57	0.92	0.48
Palliative Care Support	0.24 (0.11–0.54)	0.37 (0.15–0.91)	0.48

## Data Availability

Not applicable.

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
