# Peer review of "Nursing Home Residents Hospitalization at the End of Life: Experience and Predictors in Portuguese Nursing Homes"

_ijerph, 2023, doi:10.3390/ijerph20020947_

Round 1
Reviewer 1 Report
This manuscript is well written and adds to the body of knowledge about care-giving in nursing homes. It requires very few grammatical corrections, such as the following example of lack of concord between subject and verb:
Some of the possible consequences of the lack of appropriate healthcare coverage is the under-recognition of palliative care needs

Author Response
The authors thank the reviewer for the thorough analysis, and the pertinent comments, that contributed to improve the manuscript. We are pleased to address the reviewer comments and submit a revised version of our paper. In the attached file we provide a point-by-point response to the comments.

Reviewer 2 Report
Dear authors,
thank you very much for this very fine study concerning deceased NHR and the amount of NHR who experienced transition to an emergency room/hospital before death.
It is worth publishing, there are two minor corrections that have to be performed:
1) Page 3, Results, line 127: The study included a sample of 7 NH. How were these 7 NH chosen? This remains unclear, please explain and add.
2) Page 3, line 100 and Table 1: chronic health diagnosis. How did you find out? From discharge letters, or questionaire for the GPs? And how was the syndrome dementia evaluated? Just from a medical record or by testing via MMSE e.g.?
Author Response

(The authors gave the same response as above.)

Reviewer 3 Report
First of all, I would like to thank the authors for the pleasant study and the relevance of the topic. However, I need some clarifications, as the topic is still being explored in the literature.
Background:
Line 28-29: please, better define the problem of ageing and frail elderly with more recent data and more citations:
Line 51-55: please, better define the topic “burdensome transition”, as its operational definition will guide both the construction of the objective and the definition of the measured outcomes (line 106-109)
Line 60: please, define acronyms used the first time they are named
Aim:
State specific objectives, including any prespecified hypotheses, on the basis of what is defined in the background and what will be measured according to the materials and methods section (give greater linearity)
Materials and Methods
Line 92-93: please, give the rationale for the choice of cases (is there literature to support this specific choice?)
Line 106-109: It would be better to clarify the link with the study by Gozalo et al. 2011, perhaps by reporting the outcomes analysed in the background with greater clarity, in association with the definition given to burdensome transition. Is this supported by other and more recent literature?
Line 113: Was it not possible to adopt a database? This could be a bias in terms of potentially vitiated information
Variables: Clearly define all outcomes, exposures, predictors, potential confounders, and effect modifiers. Give diagnostic criteria, if applicable. For each variable of interest, give sources of data and details of methods of assessment (measurement) and any potential sources of bias (see Strobe statement)
Results
Report numbers of individuals at each stage of study—eg numbers potentially eligible, examined for eligibility, confirmed eligible, included in the study, completing follow-up, and analysed. Give reasons for non-participation at each stage (from 206 to 176?)
The results are detailed but lack some information, e.g., on reported hospitalisation for urinary tract infections or sepsis (referred to line 109)
Discussion and Conclusion
Line 280-285: The fact of being younger, i.e. under 85 correlates with a higher risk of being transferred to hospital and, I assume, indirectly of death. I would delve deeper into the issues underlying this with the available literature. Is it possible that a transfer to hospital is more commonly decided for a younger person as such? Could the fact that the specialised personnel veered towards the choice of transfer be attributable to a more specialised assessment of the patient?
Author Response

(The authors gave the same response as above.)
